# Spelling performance on the web and in the lab

**Arnaud Rey** [1,2]*, **Jean-Luc Manguin**[3], **Chloé Olivier**[1], **Sébastien Pacton**[4], **Pierre Courrieu**[1]

**1** Laboratoire de Psychologie Cognitive, CNRS—Aix-Marseille Université, Marseille, France, **2** Institute of Language, Communication and the Brain, Aix-Marseille Université, Marseille, France, **3** GREYC, CNRS—Université de Caen Basse-Normandie–ENSICAEN, Caen, France, **4** Laboratoire Mémoire, Cerveau et Cognition, Université Paris Descartes, Paris, France

* arnaud.rey@univ-amu.fr

**Data Availability Statement:** Data and materials for all the experiments are available on Open Science Framework (Doi: 10.17605/OSF.IO/XFQ96).

**Funding:** AR was supported by the BLRI Labex (ANR-11-LABX-0036), the Institut Convergence

## Abstract

Several dictionary websites are available on the web to access semantic, synonymous, or spelling information about a given word. During nine years, we systematically recorded all the entered letter sequences from a French web dictionary. A total of 200 million ortho-graphic forms were obtained allowing us to create a large-scale database of spelling errors that could inform psychological theories about spelling processes. To check the reliability of this big data methodology, we selected from this database a sample of 100 frequently mis-spelled words. A group of 100 French university students had to perform a spelling-to-dicta-tion test on this list of words. The results showed a strong correlation between the two data sets on the frequencies of produced spellings (r = 0.82). Although the distributions of spell-ing errors were relatively consistent across the two databases, the proportion of correct responses revealed significant differences. Regression analyses allowed us to generate possible explanations for these differences in terms of task-dependent factors. We argue that comparing the results of these large-scale databases with those of standard and con-trolled experimental paradigms is certainly a good way to determine the conditions under which this big data methodology can be adequately used for informing psychological theories.

## Introduction

Due to the exponential growth of computer capacities both in terms of storage and processing, a new area of research has emerged over the last decade, notably in the field of psychology. Frequently coined the "big data" (r)evolution [1], it simply reflects a new scientific situation that allows us to collect, store and access massive amounts of data that measure various aspects of human behaviors. One of the main concerns for experimental psychologists in front of these new gigantesque datasets is to determine how to use them in order to adequately inform psychological theories. The present study is precisely designed to address this issue by considering a new large-scale database of spelling performances that has been collected over years from a French dictionary website.

ILCB (ANR-16-CONV-0002), and by the CHUNKED
ANR project (#ANR-17-CE28-0013-02). SP was
supported by grants from Université Paris
Descartes and by the Labex "Empirical
Foundations of Linguistics" (ANR-10-LABX-0083).
The funders had no role in study design, data
collection and analysis, decision to publish, or
preparation of the manuscript.

**Competing interests:** The authors have declared
that no competing interests exist.

Collecting large amounts of data from the web is no longer a problem for the very last generations of computers and for researchers studying the cognitive processes involved in written production. These new databases can even provide strong empirical constraints for testing computational models of written word production [2]. For example, it would be useful to know the set of possible spelling errors that can be produced for a given word and, more precisely, to have access to robust estimates of the quantitative distribution of these errors. Let us, for example, consider the English word "inaccurate" and the following spelling error "innacurate" [3]. Can we get a good estimate of the proportion of that misspelling among all erroneous spellings related to "inaccurate" and can we provide a theoretical account for this proportion in terms of computational models and predictions generated from computer simulations? In the present study, on the basis of millions of web entries, we were able to connect most erroneous productions (i.e., letter strings that do not appear as words in a lexical database) to their base word and to compute estimates of the distribution of spelling errors for a large set of words.

Developing new methodologies that increase the grain size of empirical data frequently help increasing the grain size of models and theories. In the related domain of visual word recognition, large-scale item-level databases have been introduced two decades ago to test the descriptive adequacy of various existing models and this approach has been particularly useful in pushing forward the precision of both data and theories [4–9]. However, large-scale reading studies can only be done in the lab because they require recording devices for carefully measuring the main dependent variable, i.e., response times. In the best cases, 30 participants produced naming times for a set of 3,000 words, leading to a dataset of 90,000 datapoints [6].

In the case of spelling, one important limitation of lab experiments is the restricted number of items participants can process. Obtaining a database including at least 1,000 words would require a minimum of 3 hours per participant and although this spelling-to-dictation test could be done in several separated sessions, the quality of spelling patterns after one hour of experiment would certainly be affected by fatigue and boredom. Collecting spelling patterns on the Web may provide a solution to these limitations. Indeed, the scale of magnitude regarding the number of items composing the database can be far larger with no restriction on a specific set of words. It can also rapidly be applied to several languages having different phonographic structures with no need to run the same resource consuming experiments. Large-scale databases collected on the web could then open a new era for cognitive modeling in the domain of spelling production.

A critical difficulty with the data collected on the Web is that, unlike standard experimental data collected under controlled experimental conditions and following a set of precise instructions, there is virtually no information on how these spellings been produced. Although the big data methodology can provide robust–and almost noise-free–empirical estimates of spelling performances due to the huge number of collected data points, there could be qualitative differences between these large datasets and standard experimental data collected in the lab due to unexpected biases or procedural differences. That is the reason why comparing the results of both methodologies (i.e., big data *vs.* standard experimental data from the lab) should allow us to check whether spelling performances extracted from a big data methodology do display the same quantitative and qualitative properties than the experimental data generated from the lab.

In the present study, spelling errors were collected in French, a language that is particularly inconsistent when one needs to retrieve the orthographic transcription of a spoken word. Let us first briefly review three of the main characteristics of the French orthography that explain why misspellings are rather frequent in this language, even in highly educated adults. First, like many other languages with an alphabetic system, French has an inconsistent one-to-one

mapping between phonemes and graphemes. Using computer simulations, it has been shown that the application of sound-to-spelling rules allows for the correct spelling of no more than one half of all French words [10]. This is largely because there is often more than one spelling for a phoneme [11–13]. For example, /o/ can be spelled *o*, *au*, *eau*, *ot* and /ã/ can be spelled *an*, *en*, *ant*. A second characteristic of French is that spellers often have to choose between single-letter and double-letter spellings for consonant phonemes. For example, /f/ is spelled as *f* in *moufle* (mitten) and *ff* in *souffle* (breath), and French spellers sometimes omit a doublet, misspelling *souffle* as *soufle*, and sometimes erroneously double a letter, misspelling *moufle* as *mouffle* [14]. A third characteristic is that many letters are silent or do not have any phonological counterparts [15,16]. For example, the final *d* of the words *bavard* (talkative) and *foulard* (scarf) are not pronounced. Similarly, the plural markers–*s* and–*nt* are also silent: "*elle danse*" (she dances) and "*elles dansent*" (they dance) do have exactly the same pronunciation /ɛl dã:s/. In order to spell French correctly, one must therefore acquire and deploy several linguistic abilities, making use of lexical, morphological and morpho-syntaxic information that go far beyond the sound-to-spelling transcription rules [17].

Over the last years, there have been extensive efforts to develop dictionary web sites freely providing information about words (e.g., definition, etymology, synonyms). That is the case, for example, for the electronic dictionary of synonyms from CRISCO [18], which was used in the present study. For nine years, one of us (JLM) systematically recorded all the requests that were addressed by web users to the dictionary of synonyms. About 200 million orthographic forms were collected and used to create a large-scale database of spelling performances including both correct orthographic forms and errors.

However, as mentioned above, it is unclear whether the same kind of spelling errors is produced with a dictionary website and in other situations such as spelling to dictation, or typing words in sentence contexts, for instance. Task constraints are probably not the same, and we cannot be sure, a priori, that spelling errors will be similar and will have the same distribution in various tasks. For example, [19] compared the frequencies of occurrence of 351 correct or misspelled forms obtained from a dictionary website (isolated words) and from discussion forum websites (words in sentence context) and a correlation of r = 0.76 (p<0.0001) was found between the form log-frequencies of the two data sets. Although strong and significant, this correlation is not perfect, and one cannot guaranty that there is no systematic or qualitative difference between the two distributions of spelling performances.

In fact, in every set of item means (letter string frequencies in the present case), there is a part of the item variance that is systematic (the "item effect"), and a part that is random (noise). One can consider that two sets of item means belong to the same data population only if their correlation account for the systematic part of the item variance. A method for estimating this systematic item variance part has been proposed recently and it is based on a particular intraclass correlation coefficient (ICC) that will be used in the present study [4,5,7,8].

If one observes a reasonable agreement between the spelling data obtained from a dictionary website and those obtained in another situation, say in spelling to dictation, then it will be possible to generalize the observations made on automatically generated large-scale databases of spelling errors to other common situations. However, if some systematic difference appears, then we must take this difference into account when analyzing a set of spelling productions in order to adequately use these large-scale databases to model written word production processes.

To address this issue, a spelling to dictation experiment was conducted on a large sample of participants (i.e., 100) with words selected from the website database. The resulting data were compared to those obtained with the dictionary website in order to estimate the similarities and possible qualitative differences between the two datasets.

## The web-dictionary database

The observation corpus originated from all requests to the electronic dictionary of synonyms from CRISCO [18] during the first 9 years it was put on-line (October 1998—December 2007). This corpus corresponds to about 200 million requests, and about 4 million distinct orthographic forms were observed. Only those appearing more than 200 times were selected resulting in a set of 58.509 orthographic forms.

From this large dataset, orthographic forms could correspond either to a word or a misspelled word. The reference lexicon used to retrieve the French words corresponding to the letter strings that were typed by users was the MORPHALOU French open morphological lexicon (about 540.000 lexical entries; [20,21]). Whenever the entered letter string was found in the reference lexicon, it was classified as a correct spelling. Among the set of 58.509 orthographic forms, 43.444 were correct spellings (i.e., 74.25%). Otherwise, an approximate string-matching procedure was applied to the erroneous letter string in order to retrieve its associated lexical entry. This procedure was used for the resulting 15.065 erroneous letter strings.

This procedure was composed of 3 steps. First, all diacritic marks were ignored since users frequently omit or mistype them. If an entered string matched a lexical entry (by ignoring all diacritic marks), then the string was associated to the lexical entry. For example, the erroneous string "*abime*" was associated to the lexical entry "*abîme*" (abyss). Second, a list of orthographic neighbors of the entered string was generated from the lexicon using a Levenshtein-Damerau distance of 1 [22]. Orthographic neighbors were obtained by insertion, deletion, or substitution of a single character, or a transposition of two adjacent characters. If a lexical entry was obtained after one of these transformations, then the string was associated to the lexical entry. For example, the erroneous string "*acceil*" was associated to the lexical entry "*accueil*" (reception) that corresponds to an orthographic neighbor obtained by the transposition of "eu" and "ue". Third, a phonological form of the entered string was generated using grapheme-to-phoneme correspondences. If the resulting phonological form matched the phonological form of a lexical entry, then the string was associated to the lexical entry. For example, the erroneous string "*aluciner*" was associated to the lexical entry "*halluciner*" (hallucinate) because both shared the same phonological form (i.e., /alysine/).

By using this 3-steps procedure, 12.946 (85.9%) associations between an erroneous letter string and a lexical entry were automatically generated. The remaining 2.119 strings were hand-coded (530) or dismissed (1589) when no related lexical entry could be identified. For example, the entered string "*appuier*" was not associated by the 3-steps procedure to the lexical entry "*appuyer*" (to press) and was therefore hand-coded. Alternatively, the entered string "*ajeun*" was dismissed because it did not match any single-word lexical entry ("ajeun" being related to the expression "*à jeun*"–on an empty stomach—that is composed of two distinct words).

## Materials and methods

A sample of 100 words whose percentage of spelling errors in the website database varied from 3.81% to 79.64% (average 29.16%) was randomly selected from the database. The number of occurrences of these words (word requests with or without spelling error) in the website database varied from 947 to 79818, and their frequency varied from 0.07 to 236.89 occurrences per million according to the "Lexique 3" count in books [23]. We found a positive correlation between the number of occurrences in the database and the log-frequency of words (r = 0.39, p<0.0001).

### Participants

A group of 100 French native speakers, university students (68 females and 32 males, 23.25 years old on the average, s.d. = 2.96) participated in the experiment. For the present

experimental procedure (a simple spelling-to-dictation test), no formal approval was required from our institutional or national ethic committee. Written informed consents from participants were recorded.

## Procedure

The participants had to perform a spelling-to-dictation test on the selected list of 100 words. Before the dictation, each participant received a sheet of paper with a grid of 100 numbered cells. Then the dictation of the 100 words began, and the participants had to write each word by hand in the appropriate cell (in increasing order). The dictation duration was about 20 minutes (12 seconds per word). The produced letter strings were then entered in a computer program for the analysis.

## Results

A total of 653 distinct misspelled strings were obtained, in addition to the 100 correct words, in the database or in the experiment. 593 misspelled strings appeared in both data sets, 29 misspelled strings appearing only in the website database and 31 only in the dictation experiment. All 753 appearing strings (i.e., 653 misspellings and 100 correct spellings) were taken into account in the analyses. For each string, a frequency of occurrence was computed for the website and the dictation databases in order to compare these databases at the item level. Concerning the website database, the frequency of each string was the ratio of its number of occurrences on the number of occurrences of all strings related to the target word (multiplied by 100). For the dictation data, a data table of 753 strings-by-100 participants was built, with the value 1 for each cell where the participant produced the string, and the value 0 otherwise. The string frequencies are just equal to the item sums of this table or to the total number of participants that have produced that string. Note that strings not appearing in a given data set had a zero frequency in that data set. Table 1 provides an example with the target word "*hallucinant*" (hallucinating) and its related erroneous strings.

### Amount of systematic item variance

To estimate the reliability and robustness of the string frequencies obtained in the dictation data (and respectively, the amount of experimental noise), one can determine the amount of systematic variance that is present in this dataset. Practically, suppose that the same dictation

**Table 1. Frequencies computed for the website and dictation databases for the target word "*hallucinant*" (hallucinating).**

| target word | occurrences in the website database | orthographic strings | response type | frequency in the website database | frequency in the dictation |
|---|---|---|---|---|---|
| *hallucinant* | 3510 | hallucinant | correct | 79.9 | 79 |
| | 417 | allucinant | error | 9.5 | 16 |
| | 221 | alucinant | error | 5,0 | 0 |
| | 244 | halucinant | error | 5.6 | 3 |
| | 0 | allusinant | error | 0 | 1 |
| | 0 | hallucinent | error | 0 | 1 |
| | *Total = 4392* | | | | |

For each string, the website frequency is computed by dividing the number of occurrences obtained for that string by the total number of strings related to the target word (i.e., 4392). The dictation frequency is simply equal to the number of participants having produced that string. Note that two strings were not present in the website database (i.e., "allusinant" and "hallucinent") but were produced in the dictation experiment. Conversely, one string appeared in the website database (i.e., "alucinant") and not in the dictation experiment.

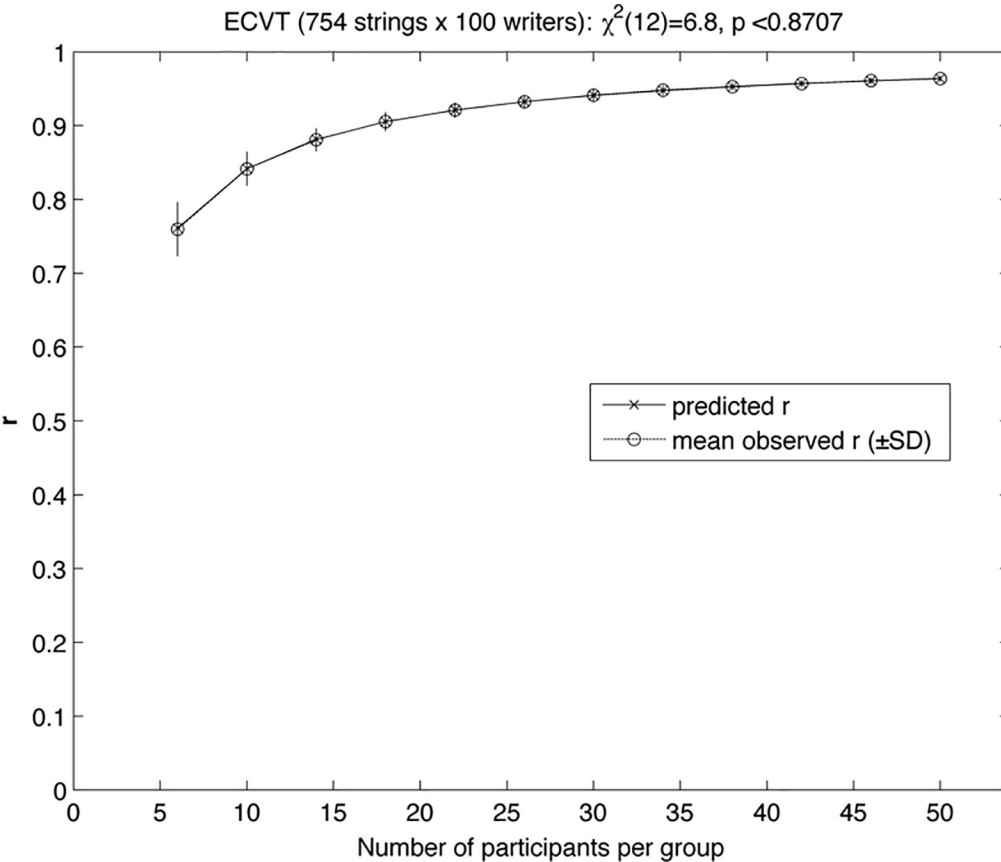

**Fig 1. ECVT test for the production frequency of correct and misspelled strings under word dictation.**

test was done on an independent sample of 100 different participants under the same experimental conditions, estimating the amount of systematic variance should tell us the range of correlations that should be obtained between the string frequencies of these two independent groups of 100 participants. If the level of experimental noise is low then the correlation between the two groups and the amount of shared systematic variance should be high, indicating that the resulting item means are robust estimates of item performances.

It has been shown that the proportion of systematic variance available in the item means of an (m items)-by-(n participants) data table can be suitably estimated using an Intraclass Correlation Coefficient (namely the "ICC(2, k)", according to the nomenclature of [24], provided that the experimental measure follows an additive decomposition model [7,8]. This last condition can easily be tested using the "Expected Correlation Validation Test" (ECVT) proposed in [8]. This test was applied to the 753 strings-by-100 writers data table from the dictation experiment (i.e., m = 753 and n = 100). As one can see in Fig 1, there was no visible or significant difference between the theoretical model prediction from the ECVT test and the empirical correlation function, which means that we can confidently use the ICC statistic to estimate the proportion of systematic variance in the string frequencies from the dictation data. The ICC of these data is equal to 0.9813, with a 99% confidence interval of [0.9787, 0.9837], that is, there is about 98% systematic variance in the vector of 753 string frequencies indicating that the dictation database is highly reliable and provides robust estimates of spelling error distributions.

## Comparison between the web database and the dictation data

The correlation coefficient between the production frequencies of the 753 observed strings in the database and under dictation is r = 0.8191, that is, the two data sets share about 67.1% item variance (if one considers that the website string frequencies are approximately noise-free due to the large number of observations). This percentage of shared variance is much less than expected from the ICC (98.13%). Thus, it is clear that there is some systematic difference between these two data sets. The correlation between the frequencies of correct spellings was r = 0.6175 (N = 100), while the correlation between the frequencies of spelling errors was r = 0.7588 (N = 653). Thus it seems that the main discrepancy between the two data sets concerns the frequencies of correct spellings.

A critical step in this study was the string-matching procedure that was applied to erroneous letter strings collected from the Web Dictionary to retrieve their associated lexical entries. Applying this procedure allowed us to automatically recover a large proportion of base words (i.e., 85.9%). The lab experiment can also inform us about the validity of this procedure because the base words are known, by definition, in the spelling test under dictation. We therefore applied (thanks to a judicious suggestion from one of the reviewers) the same string-matching procedure to the spelling errors collected during the dictation test. In the same way, we found that 90.4% of these errors could be related to the correct base word by applying this automatic procedure. This result indicates that we can be confident using this procedure to retrieve the associated lexical entries, which is crucial for building the database.

## Quantitative differences

In the website database, the average percentage of correct word spellings is 70.84%, while for the word dictation, the percentage of correct word spellings is 48.03%. In order to test this difference, we built an "accuracy regressor" in the following way: a coefficient equal to 1 was associated to each correct spelling (100 strings), and a coefficient equal to -1 to each misspelling (653 strings). The correlation between the accuracy regressor and the website database string frequency was r = 0.9305, while this correlation was only r = 0.6763 for the dictation string frequency. These two correlations are significantly different according to Williams $T_2$ test [25,26]: $T_2(750)$ = -33.7335, p<0.0001. Thus, it is clear that the productions in the database are overall much more accurate than those obtained under dictation.

## Qualitative differences

To better understand the qualitative differences between these two databases, we compared the results of two regression analyses involving different sets of regressors that are known to affect spelling performances in French [27]. In the first analysis, we tested the effect of six regressors that could be easily computed and used on both the percentage of correct spellings and the distribution of errors for both the website and the dictation databases. In the second analysis, we tested the effect of another set of 8 regressors on the percentage of correct spellings only and we compared the results of this regression analysis to the one reported in [27]. We restricted this analysis to correct spelling because the item values for these regressors were directly available from the MANULEX database [28], which was not the case for errors. This also allowed us to run the regression analysis on a larger number of items and to compare the results to another independent database. Note that the purpose of these analyses is not to provide an exhaustive account of the distribution of spelling performances but to get a first qualitative overview of the variables that are affecting performances in both databases in order to better understand the quantitative differences reported above.

## Regression analysis 1

In this first regression analysis, we used lexical and sublexical variables in order to estimate the respective contribution of these factors to spelling performances. Among the 6 regressors, there were 3 lexical variables that were extracted from the Lexique 3 database [23] and that correspond to standard variables that are known to affect spelling performances [27]. As mentioned above, since our goal was not to provide a full account of spelling performances, we used sublexical variables that were easily accessible, such as the number of letters in a word or bigram frequency counts, in order to obtain a first overview of the contribution of these variables to spelling performances in both databases. The 6 regressors were:

- The logarithm of the target word frequency (plus 1) in books (from Lexique 3). Note that 1 was added to the observed frequency in order to avoid log(0) and large negative logarithms for very rare words. This is a lexical level variable reflecting the frequency of people exposure to the target word during reading.

- The number of orthographic neighbors of the target word (from Lexique 3). This number corresponds to the number of words that can be obtained by changing one letter from the target word [29]

- The number of phonological neighbors of the target word (from Lexique 3). As for orthographic neighbors, this number corresponds to the number of words that can be obtained by changing one phoneme from the phonemic transcription of the target word.

- The string length, that is, the number of letters of the target word.

- The log frequency of the less frequent bigram in the string (from Lexique 3). This sublexical variable provides information about the frequency of the sublexical spelling patterns that are composing a word. Previous studies have indeed found that low frequency spelling patterns are more likely to be misspelled and replaced by more frequent ones [27].

- The increase of the log frequency of the less frequent bigram in misspelled strings with respect to that of the target word. This is the log frequency of the less frequent bigram in the misspelled string minus the log frequency of the less frequent bigram in the target word (thus this equals zero for all correct words). This regressor applies only to misspellings and has been shown to affect spelling performances [27].

The correlations between the six regressors are reported in Table 2.

Correlation coefficients between these regressors and the observed frequency of strings are reported in Table 3, for the website database and the dictation data, and for correct spellings and misspellings separately.

In what concerns correct spellings, the only regressor having a significant effect is the target word log-frequency, which not surprisingly increases the frequency of correct responses for both the web database and the dictation data. However, the correlation between the word log-frequency and the frequency of correct spellings is significantly lower in the database (0.235) than in the dictation data (0.4767), according to Williams $T_2$ test: $T_2(97) = 3.0951$, $p<0.003$. Another difference between the website and the dictation databases on correct spelling frequencies was observed in the correlations with string length. These correlations were not significant, however, they were in opposite directions and their difference was significant according to Williams $T_2$ test: $T_2(97) = 2.01$, $p<0.05$. Consistent with previous observations [27] in the dictation data, word length tends to have a negative effect on the frequency of correct spellings (r = -0.1039, i.e., there were more correct spellings on short than on long words). However, surprisingly, in the website database, this effect tends to be positive (r = 0.071).

**Table 2. Correlations coefficients between the six regressors for words and for misspellings.**

|  | Word frequency | Orthogr. neighbors | Phonol. neighbors | Min bigram frequency | Min big. frq. increase | String length |
|---|---|---|---|---|---|---|
| Words (N = 100) |  |  |  |  |  |  |
| Word frequency | - | 0.1259 | 0.0690 | 0.1427 | - | -0.0860 |
| Ortho. N | 0.1259 | - | 0.6160 | 0.2953 | - | -0.4334 |
| Phono. N | 0.0690 | 0.6160 | - | 0.2493 | - | -0.6463 |
| Min bigram freq. | 0.1427 | 0.2953 | 0.2493 | - | - | -0.0507 |
| String length | -0.0860 | -0.4334 | -0.6463 | -0.0507 | - | - |
| Errors (N = 653) |  |  |  |  |  |  |
| Word frequency | - | 0.1547 | 0.1318 | 0.0519 | -0.0550 | -0.1253 |
| Ortho. N | 0.1547 | - | 0.5536 | 0.1333 | -0.1014 | -0.3765 |
| Phono. N | 0.1318 | 0.5536 | - | 0.1161 | -0.0969 | -0.6321 |
| Min bigram freq. | 0.0519 | 0.1333 | 0.1161 | - | 0.5680 | -0.0619 |
| Min big. frq. inc. | -0.0550 | -0.1014 | -0.0969 | 0.5680 | - | -0.0986 |
| String length | -0.1253 | -0.3765 | -0.6321 | -0.0619 | -0.0986 | - |

Although this effect itself is not significant, the significant difference between the website and the dictation data on this point certainly requires an explanation.

In what concerns misspellings, we observed significant positive correlations between the observed string frequency and the increase of the log-frequency of the less frequent bigram, for both the database and the dictation data. In other words, misspellings tend to replace infrequent bigrams with more frequent ones, this being true in the two data sets. We also found a significant negative correlation for the website database between string length and the observed string frequency, showing that the shortest misspellings tend to be more frequent than the longest ones. No other significant effect was observed for misspellings, and globally, no qualitative difference was detected between the two data sets in what concerns misspellings.

To summarize, with the 6 regressors used in this first regression analysis, we found that observed spelling errors led globally to the same regression patterns in the two data sets. In both cases, misspellings tended to replace the less frequent bigrams of the target word with more frequent bigrams, a result that is consistent with those obtained in previous studies [27]. Thus, on the basis of this first analysis, spelling errors and their distributions look reasonably similar in the website and in the dictation data, except that misspellings are globally less frequent in the website data.

Conversely, spelling performances were significantly more accurate in the website (about 71% correct) than in the dictation database (about 48% correct). Not surprisingly, in both data

**Table 3. Correlation coefficients of the six regressors with the frequency of observed strings in the website database and in the spelling to dictation data.**

|  | Word frequency | Orthogr. neighbors | Phonol. neighbors | Min bigram frequency | Min big. frq. increase | String length |
|---|---|---|---|---|---|---|
| Correct (N = 100) |  |  |  |  |  |  |
| Web database | 0.2350* | 0.0465 | -0.0597 | 0.1341 | - | 0.0710 |
| Dictation | 0.4767*** | 0.0972 | 0.0359 | 0.0273 | - | -0.1039 |
| Errors (N = 653) |  |  |  |  |  |  |
| Web database | 0.0047 | 0.0475 | 0.0555 | 0.0362 | 0.0938* | -0.0951* |
| Dictation | -0.0727 | 0.0338 | 0.0273 | 0.0745 | 0.1029** | -0.0572 |

*: p < .05

**: p < .01

***: p < .001

sets, the correct spelling frequency increases with word frequency. However, this relation is significantly weaker in the website database. Moreover, although the word length effect in correct spelling frequencies was not significant, the correlations between word length and correct spelling frequencies were significantly different between the two data sets. These discrepancies can certainly be explained by considering the situations leading a user to consult a dictionary website.

We can indeed identify two main situations in which a user searches for the synonym of a word. In the first case, the user can have an approximate knowledge of a word that she/he planes to write in a document or to replace by a synonym. This situation is in some way comparable to the word dictation situation. However, there is a second kind of situation, where the user just encountered a word that she/he does not precisely know in a document, and she/he looks for synonyms. In this last case, the user does not have to search for information in her/his mental lexicon, but she/he has just to copy the encountered word in the electronic dictionary input window. As a result, a spelling error is less probable because the correct word was just seen, and there is therefore no reason that the produced spelling depends on the word frequency. Now, when copying an available word from a document to the dictionary input window, one can type it, with a non-zero probability of misspelling, or one can "copy and paste" it, with a zero probability of misspelling. If the word is short, then it is probably faster to type it, however, if it is a long word, one will probably prefer to copy and paste it. As a result, the probability of a spelling error tends to decrease (thus the accuracy increases) as word length increases, contrarily to what logically happens in other situations. If this is actually the case, it should be possible to detect a significant positive correlation between word length and spelling accuracy in a large sample of words from the website database. This issue will be considered in the next regression analysis.

### Regression analysis 2

While the correspondence between spelling errors obtained from the dictionary website and those obtained in the spelling to dictation experiment seems generally good, systematic differences appeared between the two data sets on the frequencies of correct spellings. In particular, the difference on the word length effect provides interesting clues to understand the observed discrepancies. Studying this effect (i.e., a positive correlation between word length and spelling accuracy) on a larger sample of words is therefore critical in order to test its reliability and to better characterize the task-dependent differences between the website and the dictation databases.

In a previous large-scale study based on the spelling production of children, Lété et al. (2008) have identified various factors influencing spelling accuracy. Although children are not necessarily representative of the general population, it is of interest to see if some of the observed effects can be reproduced using the data from the dictionary website. As for the dictation experiment, comparing the patterns of correlations between the website database and previously reported results can also help us determining how to use the new big data database adequately in order to inform psychological theories of spelling production.

To examine these questions, we selected a sample of 6567 words common to the website database and to the French MANULEX database [28], which provides a number of word characteristics that we are going to use as regressors hereafter. We selected a set of 8 regressors that were used in the study by [27] or that were available from the MANULEX database:

- WF: Logarithm of the word frequency (plus 1) per million in books (from Lexique 3). The word frequency positively influenced children's spelling accuracy in [27], as well as in the present study.

- Len: Number of letters of the word. The word length negatively influenced children's spelling accuracy in [27], as in the present spelling to dictation experiment (although the correlation was not significant in the dictation experiment). The reverse (i.e., a positive correlation) was observed for the website database (the correlation being also statistically non-significant at p = .05).

- CGP: Grapheme to phoneme correspondence—minimum consistency (from MANULEX). This is the consistency of the less consistent grapheme to phoneme correspondence in the word. This type of regressor had no clear effect on children's spelling accuracy in [27].

- CPG: Phoneme to grapheme correspondence—minimum consistency (from MANULEX). This is the consistency of the less consistent phoneme to grapheme correspondence in the word. This type of regressor positively influenced children's spelling accuracy in [27].

- HPn: Number of heterographic homophones (from MANULEX). This is a kind of phoneme to grapheme correspondence inconsistency at the word level. So, one can expect a negative effect on spelling accuracy.

- HPf: Logarithm of the mean frequency (plus 1) of the heterographic homophones (from MANULEX). This could reinforce the negative effect of HPn.

- PGNn: Number of phonographic neighbors (from MANULEX). Phonographic neighbors are simultaneously orthographic and phonological neighbors. This regressor had a positive influence on children's spelling accuracy in [27].

- PGNf: Logarithm of the mean frequency (plus 1) of the phonographic neighbors (from MANULEX). This regressor had a negative influence on children's spelling accuracy in [27].

The global descriptive statistics of the sample of 6567 words are shown in Table 4.

Table 5 shows the correlations between all regressors and the spelling accuracy in the website data. As expected, we found an effect of the WF and CPG regressors and no significant effect of the CGP regressor. However, unlike [27], no significant effect of PGNn and PGNf was detected, and the effect of word length (Len) was significantly positive instead of negative. This last result seems to confirm the hypothesis that, for users of the website, the target word is in fact available in a non negligible proportion of cases, and it can be entered in the search engine by typing (with possible errors) or by copy and paste (without error), with a greater probability of using copy and paste for the longest words than for the shortest ones. Finally, one observes significant effects of the HPn and HPf regressors, but in the opposite direction of what was

**Table 4. Statistics of the sample of 6567 words from the website database.**

|  | min | max | mean | sd |
|---|---|---|---|---|
| Num. of occurrences in the website database | 200 | 290554 | 9326 | 15576 |
| Spelling accuracy | 0.1176 | 0.9988 | 0.7835 | 0.1619 |
| Frequency per million in books (from Lexique 3) | 0 | 5186.8 | 17.6862 | 98.7846 |
| Number of letters | 3 | 18 | 8.2164 | 2.1936 |
| From MANULEX: |  |  |  |  |
| Grapheme to phoneme minimum consistency | 0 | 100 | 41.6152 | 24.7274 |
| Phoneme to grapheme minimum consistency | 0 | 100 | 23.1678 | 19.3699 |
| Num. of heterographic homophones | 0 | 13 | 1.3224 | 1.8886 |
| Freq. of heterographic homophones | 0 | 1053 | 3.6534 | 21.6155 |
| Num. of phonographic neighbours | 0 | 9 | 0.4570 | 0.9104 |
| Freq. of phonographic neighbourhood | 0 | 3347.5 | 4.3722 | 52.4606 |

**Table 5. Correlation matrix of the spelling accuracy in the website database and the 8 tested regressors.**

|  | **Accu** | **CGP** | **CPG** | **WF** | **Len** | **HPn** | **HPf** | **PGNn** | **PGNf** |
|---|---|---|---|---|---|---|---|---|---|
| Accu | - | -0.012 | 0.135 | 0.242 | 0.120 | 0.063 | 0.026 | 0.011 | 0.016 |
| CGP | -0.012 | - | 0.236 | 0.031 | -0.298 | 0.096 | 0.001 | 0.140 | 0.112 |
| CPG | 0.135 | 0.236 | - | 0.028 | -0.174 | 0.065 | -0.004 | 0.098 | 0.079 |
| WF | 0.242 | 0.031 | 0.028 | - | -0.204 | 0.215 | 0.429 | 0.204 | 0.217 |
| Len | 0.120 | -0.298 | -0.174 | -0.204 | - | -0.231 | -0.232 | -0.355 | -0.300 |
| HPn | 0.063 | 0.096 | 0.065 | 0.215 | -0.231 | - | 0.396 | 0.332 | 0.180 |
| HPf | 0.026 | 0.001 | -0.004 | 0.429 | -0.232 | 0.396 | - | 0.207 | 0.223 |
| PGNn | 0.011 | 0.140 | 0.098 | 0.204 | -0.355 | 0.332 | 0.207 | - | 0.611 |
| PGNf | 0.016 | 0.112 | 0.079 | 0.217 | -0.300 | 0.180 | 0.223 | 0.611 | - |

Significance: $p < .05$: $|r| > 0.024$; $p < .01$: $|r| > 0.032$; $p < .001$: $|r| > 0.041$; N = 6567.

Accu: spelling accuracy; CGP: grapheme to phoneme min consistency; CPG: phoneme to grapheme min consistency; WF: word log-frequency; Len: number of letters; HPn: number of heterographic homophones; HPf: log-frequency of heterographic homophones; PGNn: number of phonographic neighbours; PGNf: log-frequency of phonographic neighbours.

expected. However, since there are important inter-correlations between the regressors, a hierarchical regression analysis is necessary in order to disentangle the respective role of these variables.

We used a stepwise strategy for the hierarchical regression analysis, entering the regressors in decreasing order of their correlation with the spelling accuracy, and excluding those regressors that failed to account for a significant part of the residual. The result of this analysis is presented in Table 6. As one can see, the hierarchical regression analysis essentially confirmed the observations made on the simple correlations, except that the effect of HPf is in fact significantly negative ($\beta = -0.0809$) when the effect of other regressors is taken into account. This last result brings the HPf regressor effect back to the expected negative direction, however the effect of HPn remains clearly positive. A possible explanation of the positive effect of the number of heterographic homophones results from the fact that if a user enters an heterographic homophone instead of the actual target word, then no error will be detected by the search engine since the homophone is also a word. The probability of such a situation logically increases as the number of heterographic homophones of the target word increases, hiding a number of undetected spelling errors and artificially increasing the spelling accuracy score for heterographic homophones. Note that a similar difficulty could occur in word spelling to dictation tasks, since homophones can only be distinguished by providing contextual information in the spoken language, as in [27].

**Table 6. Stepwise hierarchical multiple regression analysis of the spelling accuracy in the website database (6567 items).**

| Regressor | $R^2$ | $\Delta R^2$ | Significance | $\beta$ |
|---|---|---|---|---|
| WF | 0.0586 | 0.0586 | $F(1,6565) = 408.7$, $p < .0001$ | 0.2996* |
| CPG | 0.0749 | 0.0163 | $F(1,6564) = 115.8$, $p < .0001$ | 0.1570* |
| Len | 0.1141 | 0.0392 | $F(1,6563) = 290.5$, $p < .0001$ | 0.2055* |
| HPn | 0.1159 | 0.0018 | $F(1,6562) = 13.16$, $p < .0003$ | 0.0684* |
| HPf | 0.1205 | 0.0046 | $F(1,6561) = 34.63$, $p < .0001$ | -0.0809* |

$\beta$ significance

*: $p < 10^{-7}$.

WF: word log-frequency; CPG: phoneme to grapheme min consistency; Len: number of letters; HPn: number of heterographic homophones; HPf: log-frequency of heterographic homophones.

## Discussion

The goal of the present study was to determine if a large-scale database on spelling productions automatically collected from a dictionary website could be used to inform models and theories of written word production. By comparing the distribution of performances on correct and erroneous responses between the website database and a spelling to dictation database recorded under standard experimental conditions, we found strong similarities between the two databases regarding the distribution of errors but also significant differences regarding the proportion of correct responses, indicating the influence of task-specific factors.

Regarding the distribution of spelling errors, it seems that the website database could provide useful empirical data that are both qualitatively and quantitatively consistent with the data collected in a standard spelling to dictation test. In a regression analysis, we found that the generation of errors is indeed mainly constrained by sublexical factors and notably by the presence of low-frequency bigrams that could be replaced by higher frequency spelling patterns (see [30–34] for recent studies investigating the role of alternative factors accounting for spelling error production). Similarly, both databases produced approximately the same set of error forms (i.e., out of the 653 recorded misspellings, 593 were observed in both databases). Therefore, for errors, task-dependent factors do not seem to change drastically spelling performances and we can confidently state that the distribution of errors in the large-scale website database can be used to constrain models of written word production.

The comparison between the Web database and the lab dictation test also revealed that the two data sets share about 67.1% item variance, which is much less than expected from the ICC (98.13%). One reason that could explain this discrepancy is likely due to the way participants produced their responses. In the dictation test, words were written by hand while they were typed on a keyboard for the dictionary website. It is possible that typing encourages some types of misspellings and suppresses others in comparison with handwriting because of the physical parameters of the keyboard layout. For example, factors like the proximity of characters on a keyboard, or mono-manual *vs.* bimanual typing, could contribute to a significant part of this unshared variance. A more adequate comparison would be an experimental task where participants are given printed words and are asked to type them, rather than produce orthography based on phonology.

We also found that significant differences appeared between the two data sets on the proportion of correct responses, the number of correct spellings being larger in the website database and the correlation with word frequency being twice larger in the dictation database. Regression analyses revealed a lower correlation between correct responses and word frequency, together with a reversed correlation with word length. These results are consistent with the idea that on many occasions, website users already have access to the target word and they are simply entering it in the search engine by typing (with possible errors) or by copy and paste (without error). This initial access to the target word is likely reducing the word frequency effect that is observed in the dictation database. Similarly, there is a greater probability of using copy and paste for the longest words than for the shortest ones, therefore reversing the length effect observed in the dictation database. The second regression analysis is consistent with that hypothesis by also revealing a positive correlation between the proportion of correct responses and word length on a larger sample of words from the website database.

The effect of the number of heterographic homophones was also found to be positively related to the proportion of correct responses, contrary to what has been reported in a previous study [27]. As argued above, a possible explanation of this result also comes from the way users are entering their request on the website. Indeed, if an heterographic homophone

(instead of the target word) is entered, then no error will be detected by the search engine since the homophone is also a word. This situation artificially increases the spelling accuracy score for words having many heterographic homophones.

The present set of results therefore suggest that length effects and effects of the number of heterographic homophones must be considered with caution, since they are possibly biased in the website database by the procedure related to data collection. Due to these task-dependent factors, effects can therefore be modulated even if previously reported effects that are of particular importance in modelling spelling processes are in fact clearly reproduced. This is the case of the word frequency effect and of the phoneme-to-grapheme correspondence consistency effect, whose coexistence characterizes "dual-route" models of spelling [2]. So, as for any experimental paradigm, the use of large-scale website databases requires a fine-grained analysis of the task-dependent procedures that may generate qualitative bias in the collected data.

In the present study, we found that errors collected on the Web largely follow the distribution of errors collected in a standard spelling-to-dictation test suggesting that the same word production processes were engaged. These errors could then be used confidently to test the predictions of computational models of single word production for the entire set of existing words from a given language. Conversely, concerning the percentage of correct spellings, we now know from our analyses that word production on the Web can differ from a spelling-to-dictation test due to copy/paste procedures that will bias the resulting distribution of correct spellings. Unless we find a way to correct this bias, psychological theories of word production can therefore only benefit from this large-scale database on the distribution of errors.

Do the present results generalize to more ecological word production situations in which words are embedded within sentences and are not only produced in isolation (like during a Web request or a spelling test)? To get an answer to this question, one would certainly need to adopt a similar strategy as the one used in the present study by collecting written texts in the lab and collecting spelling productions on the Web through discussion forums, for example. A first answer comes from a comparison done by one of us [19] who compared the frequencies of occurrence of 351 correct or misspelled forms obtained from a dictionary website (isolated words) and from discussion forum websites (words in sentence context). A correlation of r = 0.76 (p<0.0001) was found between the form log-frequencies of the two data sets which is similar to the one we found in the present study. Therefore, although typing words in sentences certainly requires additional cognitive processes compared to single word production, this result suggests that the distribution of spelling errors might be quite similar and not influenced by the activation of these additional processes.

## Conclusion

The use of big data in psychology requires the same task analysis as for any study in experimental psychology in order to adequately use these massive flows of information to inform psychological theories about the structure and dynamics of mental processes. We have shown that comparing the results of these large-scale databases with the ones of standard and controlled experimental paradigms is certainly a good way to identify these task-dependent factors that any theory needs to take into account. In the present situation, while the percentage of correct responses is certainly not adequate for studying written word production processes, spelling error distributions from the large-scale internet database appears not only to be suitable to constrain models of word production at the item level but also, to provide reliable and almost noise-free observations due to the extremely large number of data points.

## Acknowledgments

We are grateful to Francesca Peressoti, Victor Kuperman and an anonymous reviewer for their helpful comments and suggestions.

## Author Contributions

**Conceptualization:** Arnaud Rey, Sébastien Pacton, Pierre Courrieu.

**Data curation:** Arnaud Rey, Jean-Luc Manguin, Chloé Olivier.

**Formal analysis:** Arnaud Rey, Jean-Luc Manguin, Pierre Courrieu.

**Funding acquisition:** Arnaud Rey.

**Investigation:** Arnaud Rey, Chloé Olivier, Pierre Courrieu.

**Methodology:** Arnaud Rey, Jean-Luc Manguin, Chloé Olivier.

**Project administration:** Arnaud Rey.

**Software:** Jean-Luc Manguin, Pierre Courrieu.

**Supervision:** Arnaud Rey.

**Validation:** Pierre Courrieu.

**Visualization:** Pierre Courrieu.

**Writing – original draft:** Arnaud Rey, Pierre Courrieu.

**Writing – review & editing:** Arnaud Rey, Sébastien Pacton, Pierre Courrieu.

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
