## [Decision Letter · Decision Letter 0]

28 Aug 2019

PONE-D-19-20392

Spelling performance on the web and in the lab

PLOS ONE

Dear Arnaud,

Thank you for submitting your manuscript to PLOS ONE. After careful consideration, we feel that it has merit but does not fully meet PLOS ONE’s publication criteria as it currently stands. Therefore, we invite you to submit a revised version of the manuscript that addresses the points raised during the review process.

In accordance with the Reviewers, I consider your work highy relevant and I think that the manuscript could be substantially improved by the suggestions you received. In particular, consider the concern of Reviewer 1 about the comparison between errors from the database and errors obtained in the experimental task. Moreover, both Reviewers agree on the fact that analogies/differences between the perfomance in the two tasks should be considered and discussed more deeply.

We would appreciate receiving your revised manuscript by Novembre 30th, 2019. To enhance the reproducibility of your results, we recommend that if applicable you deposit your laboratory protocols in protocols.io, where a protocol can be assigned its own identifier (DOI) such that it can be cited independently in the future. For instructions see: http://journals.plos.org/plosone/s/submission-guidelines#loc-laboratory-protocols

We look forward to receiving your revised manuscript.

Kind regards,

Francesca Peressotti, Ph.D

Academic Editor

PLOS ONE

Journal Requirements:

2. During your revisions, please update the title to align with PLOS ONE criteria stating that the title be specific, descriptive, concise, and comprehensible to readers outside the field.

3. Please provide additional details regarding participant consent. In the ethics statement in the Methods and online submission information, please ensure that you have specified (1) whether consent was informed and (2) what type you obtained (for instance, written or verbal, and if verbal, how it was documented and witnessed).

Additional Editor Comments (if provided):

Reviewers' comments:

Reviewer's Responses to Questions

**Comments to the Author**

1. Is the manuscript technically sound, and do the data support the conclusions?

Reviewer #1: No

Reviewer #2: Yes

2. Has the statistical analysis been performed appropriately and rigorously? 

Reviewer #1: Yes

Reviewer #2: Yes

3. Have the authors made all data underlying the findings in their manuscript fully available?

Reviewer #1: Yes

Reviewer #2: Yes

4. Is the manuscript presented in an intelligible fashion and written in standard English?

Reviewer #1: Yes

Reviewer #2: Yes

5. Review Comments to the Author

Reviewer #1: Rey and colleagues report a comparison between a spelling corpus gathered online and those gathered in the laboratory, with the goal of evaluating whether these large corpora of spelling errors are a useful tool for understanding the cognitive architecture of spelling. They find many similarities between the two data sets, but also some differences, like the effects of length and the effects of homophones.

There is a lot to recommend about this research. Figuring out how we can best use big data to inform cognitive theories is an important goal and the web seems like a remarkable tool to collect spelling errors. The systematic comparison between these web collected corpora and lab tests is an important step towards this goal. However, there are several major concerns that I have about the current manuscript that preclude it from being published in its current form.

(1) First and foremost, I am concerned about the difference in how accuracy is calculated in the two tasks. For the spelling-to-dictation task, the experimenters know the target and the response, but for the web dictionary corpora, all that is known is the response. The authors describe a procedure for linking responses to targets, though some aspects (like the hand-coding) were underspecified. I am somewhat worried about how these procedures relate to differences between two tasks. For example, heterographic homophones seem particularly challenging for this coding procedure, because writing something like THERE for “their” would be counted as a correct spelling of the target “there” instead of a heterographic homophone spelling of the target “their”. Similarly, for shorter words, it is more common for transpositions, substitutions or omissions to result in other lexical items, and therefore, by this procedure would be counted as correct. The authors acknowledge these issues to some extent, but I do not feel as if it is treated with sufficient care. A couple of suggestions I would make to address this issue more fully. a) Discuss in more detail how mapping responses to targets is one of the major challenges of using these big databases of spelling errors, as a motivation for the challenges of adopting a big data approach b) Running additional analyses with the lab-based spelling test in which the responses are analyzed using the exact same algorithm as the web-based errors, to see if the differences between the two corpora are really about task or are about coding/scoring.

(2) The theoretical need for big data approaches to spelling need to be more clearly laid out in the introduction. The authors need to make a stronger case for why corpora of lab collected spelling errors have been useful drawing conclusions about the organization of the spelling system - that is how they have made important theoretical contributions - what the limitations are of relying only on lab-based tests and what could be gained from taking a big data approach. This requires some reframing of the manuscript, but will make its contribution more strong.

(3) I agree with the statement at the beginning of the conclusion section: “The use of big data in psychology requires the same task analysis as for any study in experimental psychology in order to adequately use these massive flows of information to inform psychological theories about the structure and dynamics of mental processes.” The notion of task-related differences between the spelling-to-dictation task and the writing processes involved in looking up a word in an online dictionary is an important one. However, I think that the authors did not do a sufficient job discussing the differences between these tasks, even if it is speculative. What are the differences in mental processes between the two tasks, and how might those differences in mental processes relate to the differences observed in which variables are predictive of performance (if these differences are indeed due to differences in mental processes and not due to differences in coding)?

Reviewer #2: Review of PONE-D-19-20392 Spelling performance on the web and in the lab

The paper compares frequency distribution of correctly and incorrectly spelled words obtained from a large electronic resource (200 million tokens of data entry to an online dictionary) and a spelling-to-dictation experiment. The proposed motivation is to establish whether the big data resources are reflexive of written production behavior and whether they can be used as a basis for theoretical models of written production. The distributions of spelling errors were fairly similar in the online dataset and the experiment, while the distributions of correct spellings were substantially different (with an online dictionary showing a much larger proportion of correct spellings). The disrepancy is explored in regression models and is explained as a difference in tasks (looking up a word in the dictionary is not the same as writing it to dictation). The paper concludes that the online resource is a valid and reliable representation of spelling error distribution in spontaneous written production.

I have a positive opinion of the paper. It addresses an interesting and timely question of how useful big data collections are as representations of human behavior and what merit they have for theory-building. The paper will be of interest to the readers involved in language research and perhaps resource creation. Statistical methods are adequate and competently conducted and reported. The discussion of the literature is substantial (but see below) and the interpretation of results is thoughtful and well aligned with the findings obtained via an experiment and a corpus. My main criticism revolves around the point that the authors of the paper make repeatedly: does the discrepancy in the demands for the experimental task and for using an online dictionary affect the utility of the latter. I outline this and some other concerns below. I recommend the paper for publication in the journal pending these revisions.

Concerns.

1. Neither the use of an online dictionary nor spelling to dictation are fully representative of the cognitive and verbal demands of spontaneous written production of texts. This production is best studied using large (unedited and un-proofread) corpora of coherent written texts produced for communicative purpose. This is different from written production of individual words motivated by a lack of knowledge or confidence in an semantic, orthographic or phonological aspects of those words (online dictionary) or timed and controlled written production of individual words based on their phonology in a lab setting (spelling to dictation). I would like to see a comment in the paper about constraints on ecological validity and theoretical value that its both methodologies have as representations of written production. Ideally, I would like to see a comparison between distributions of spelling errors in written text corpora to the present distributions: I do not request it for this paper because of the scope of such task.

1b. On the same note:

"If one observes a reasonable agreement between the spelling data obtained from a

126 dictionary website and those obtained in another situation, say in spelling to dictation, then it will

127 be possible to generalize the observations made on automatically generated large-scale databases

128 of spelling errors to other common situations. However, if some systematic difference appears,

129 then we must take this difference into account when analyzing a set of spelling productions in

130 order to adequately use these large-scale databases to model written word production processes."

I agree with this statement with the only exception. The ability to generalize the dictionary spelling data to spelling-to-dictation data does not make spelling-to-dictation a "common situation" in written language use and does not guarantee direct relevance to written word production processes.

2. The experiment required participants to write the words to dictation by hand, rather than directly typing the words using a keyboard. This design decision is unlikely to cause a big discrepancy with dictionary data, but it is possible that typing encourages some types of misspellings and suppresses others in comparison with hand-writing because of the physical parameters of the keyboard layout. For instance, computational-linguistic literature on spelling errors often discusses the influence of medium-specific factors on the prevalence of errors, like the proximity of characters on a keyboard, mono-manual vs bimanual typing, i.e. factors that are influential for typing but not for hand-writing. A brief discussion of this potential source of discrepancy is necessary; also see below for a proposal to convert the experiment into a typing task.

3. A more adequate comparison would be an experimental task where participants are given printed words and are asked to type them, rather than produce orthography based on phonology. This would resolve the question that the authors are asking about the possible reasons for a much lower rate of correct spellings in the experiment compared to the database (48% vs 70%).

4. There is a decent amount of psycholinguistic literature that uses corpora to evaluate factors affecting the distributions of specific spelling errors, which may be useful as references for the present work. In some of this work, a correlation between the frequency of the word and its frequency in a correct variant is complemented by a relevant observation that the proportion of correct spelling relative to all spellings of the word actually decreases with an increase in word frequency. I'd like to see whther this tendency is observed in the dictionary data and dictation data as well. See below and references in these papers.

Schmitz, T., Chamalaun, R., & Ernestus, M. (2018). The Dutch verb-spelling paradox in social media. Linguistics in the Netherlands, 35(1), 111-124.

Bar-On, A., and Kuperman, V. (in press). Spelling errors respect morphology: A corpus study of Hebrew orthography. Reading and Writing.

5. "By using this 3-steps procedure, 12.946 164 (85.9%) associations between an erroneous letter

165 string and a lexical entry were automatically generated."

Was the automatic generation of associations checked manually? A typical procedure is to select a random subset of items and have raters attribute them to existing lexical entities: the resulting associations are then compared against automatic ones.

6. "French has an incomplete one-to-one mapping between phonemes and graphemes." -> "Incomplete" may not be a correct term here. I believe that the authors mean "consistency" or "predictability" here. For a discussion of terminology see Schmalz et al. (2015; Psychological Bulletin and Review).

6. PLOS authors have the option to publish the peer review history of their article (what does this mean?). If published, this will include your full peer review and any attached files.

Reviewer #1: No

Reviewer #2: Yes: Victor Kuperman

---

## [Author Response · Author response to Decision Letter 0]

8 Nov 2019

Our detailed responses are listed in the file "Responses to Reviewers".

---

## [Editor Report · Decision Letter 1]

12 Nov 2019

PONE-D-19-20392R1

Spelling performance on the web and in the lab

PLOS ONE

Dear Arno

I read your revised manuscript and your response to the Reviewers. Since you have almost fully and appropriately responded to the Reviewers’ concerns and modified the text accordingly, I decided not to submit the manuscript to a second round of revision.

Considering your responses, however, I found few issues that require some further work.

1. The additional analysis suggested by Reviewer 1 is a relevant validation of the procedure used for detecting errors in the web database. For this reason it cannot be placed at the end of the GD. It should be anticipated, maybe in the Result section.

-2. Point 2 Reviewer 1. You were asked to discuss in more detail to what extent corpora of lab collected spelling errors contributed to the theoretical debate, what are the limitations of these lab studies and what could be gained in using large web based databases. I agree with the Reviewer that this additional discussion would enhance the impact of your work. However, in your revised manuscript this point has only been partially developed, with reference to reading studies. I suggest to expand this point, making reference to studies using lab based databases on spelling and typing.

I  invite you to submit a revised version of the manuscript that addresses these points.

To enhance the reproducibility of your results, we recommend that if applicable you deposit your laboratory protocols in protocols.io, where a protocol can be assigned its own identifier (DOI) such that it can be cited independently in the future. For instructions see: http://journals.plos.org/plosone/s/submission-guidelines#loc-laboratory-protocols

A rebuttal letter that responds to each point raised by the academic editor . This letter should be uploaded as separate file and labeled 'Response to Editor'.A marked-up copy of your manuscript that highlights changes made to the original version. This file should be uploaded as separate file and labeled 'Revised Manuscript with Track Changes'.An unmarked version of your revised paper without tracked changes. This file should be uploaded as separate file and labeled 'Manuscript'.

We look forward to receiving your revised manuscript.

Kind regards,

Francesca Peressotti, Ph.D

Academic Editor

PLOS ONE

---

## [Author Response · Author response to Decision Letter 1]

27 Nov 2019

Regarding the additional analysis suggested by R1, it is now placed in the results section (see paragraph starting L280). Concerning your second point, we have expanded the issue of the limitations of lab experiments involving spelling studies (see paragraph starting L79). The justification for collecting Web databases is now more clearly explained.

---

## [Editor Report · Decision Letter 2]

4 Dec 2019

Spelling performance on the web and in the lab

PONE-D-19-20392R2

Dear Dr. Rey,

We are pleased to inform you that your manuscript has been judged scientifically suitable for publication and will be formally accepted for publication once it complies with all outstanding technical requirements.

With kind regards,

Francesca Peressotti, Ph.D

Academic Editor

PLOS ONE

---

## [Editor Report · Acceptance letter]

11 Dec 2019

PONE-D-19-20392R2 

Spelling performance on the web and in the lab 

Dear Dr. Rey:

I am pleased to inform you that your manuscript has been deemed suitable for publication in PLOS ONE. Congratulations! Your manuscript is now with our production department. 

With kind regards,

on behalf of

Dr. Francesca Peressotti 

Academic Editor

PLOS ONE